# HoloPart: Generative 3D Part Amodal Segmentation

**Yunhan Yang**[1]    **Yuan-Chen Guo**[2]    **Yukun Huang**[1]    **Zi-Xin Zou**[2]
**Zhipeng Yu**[2]    **Yangguang Li**[2]    **Yan-Pei Cao**[2✉]    **Xihui Liu**[1✉]

[1] The University of Hong Kong    [2] VAST

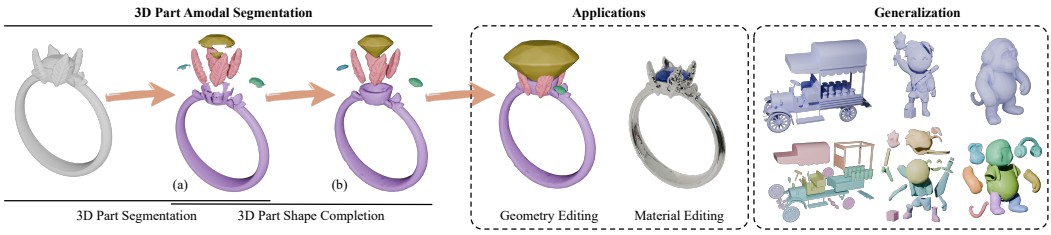

Figure 1: Demonstration of the difference between (a) 3D part segmentation and (b) 3D part amodal segmentation. 3D part amodal segmentation decomposes the 3D shape into **complete** semantic parts rather than broken surface patches, facilitating various downstream applications. In this paper, we propose a solution by performing 3D part shape completion on incomplete part segments.

## ABSTRACT

3D part amodal segmentation—decomposing a 3D shape into complete, semantically meaningful parts, even when occluded—is a challenging but crucial task for 3D content creation and understanding. Existing 3D part segmentation methods only identify visible surface patches, limiting their utility. Inspired by 2D amodal segmentation, we introduce this novel task to the 3D domain and propose a practical, two-stage approach, addressing the key challenges of inferring occluded 3D geometry, maintaining global shape consistency, and handling diverse shapes with limited training data. First, we leverage existing 3D part segmentation to obtain initial, incomplete part segments. Second, we introduce HoloPart, a novel diffusion-based model, to complete these segments into full 3D parts. HoloPart utilizes a specialized architecture with local attention to capture fine-grained part geometry and global shape context attention to ensure overall shape consistency. We introduce new benchmarks based on the ABO and PartObjaverse-Tiny datasets and demonstrate that HoloPart significantly outperforms state-of-the-art shape completion methods. By incorporating HoloPart with existing segmentation techniques, we achieve promising results on 3D part amodal segmentation, opening new avenues for applications in geometry editing, animation, and material assignment.

## 1 INTRODUCTION

3D part segmentation is an active research area. Given a 3D shape represented as a polygonal mesh or point cloud, 3D part segmentation groups its elements (vertices or points) into semantic parts. This is particularly valuable for shapes produced by photogrammetry or 3D generative models Zhang et al. (2024); Liu et al. (2023b); Hong et al. (2023); Long et al. (2024); Zhang et al. (2023); Poole et al. (2022), which are often one-piece and difficult to deal with for downstream applications. However, part segmentation has limitations. It produces surface patches rather than "complete parts" of the 3D shape like is shown in figure 1 (a), where the segmented parts are broken. This may suffice for perception tasks but falls short for content creation scenarios where *complete part geometry* is required for geometry editing, animation, and material assignment. A similar challenge has been learned in 2D for many years, through the research area of 2D amodal segmentation. Numerous

previous works Ehsani et al. (2018); Kar et al. (2015); Ke et al. (2021); Ozguroglu et al. (2024) have explored the 2D amodal segmentation task, yet there remains a lack of research for 3D shapes.

To address this, we introduce the task of **3D part amodal segmentation**. This task aims to separate a 3D shape into its *complete* semantic parts, emulating how human artists model complex 3D assets. figure 1 (b) shows the expected output of 3D part amodal segmentation, where segmented parts are complete. However, extending the concept of amodal segmentation to 3D shapes introduces significant, non-trivial complexities that cannot be directly addressed by existing 2D or 3D techniques. 3D part amodal segmentation requires: *(1) Inferring Occluded Geometry*: Accurately reconstructing the 3D geometry of parts that are partially or completely hidden. *(2) Maintaining Global Shape Consistency*: Ensuring the completed parts are geometrically and semantically consistent with the entire 3D shape. *(3) Handling Diverse Shapes and Parts*: Generalizing to a wide variety of object categories and part types, while *leveraging a limited amount of part-specific training data.*

Recognizing the inherent difficulty of end-to-end learning for this task, we propose a practical and effective two-stage approach. The first stage, *part segmentation*, has been widely studied, and we leverage an existing state-of-the-art method Yang et al. (2024) to obtain initial, incomplete part segmentations (surface patches). The second stage, and the core of our contribution, is *3D part shape completion given segmentation masks*. This is the most challenging aspect, requiring us to address the complexities outlined above. Previous 3D shape completion methods Rao et al. (2022); Chu et al. (2024); Cheng et al. (2023) focus on completing entire objects, often struggling with large missing regions or complex part structures. They also do not address the problem of completing individual parts within a larger shape while ensuring consistency with the overall structure.

We introduce **HoloPart**, a novel diffusion-based model specifically designed for 3D part shape completion. Given an incomplete part segment, HoloPart doesn't just "fill in the hole". It leverages a learned understanding of 3D shape priors to *generate* a *complete and plausible* 3D geometry, even for complex parts with significant occlusions. To achieve this, we first utilize the strong 3D generative prior learned from a large-scale dataset of general 3D shapes. We then adapt this prior to the part completion task using a curated, albeit limited, dataset of part-whole pairs, enabling effective learning despite data scarcity. Motivated by the need to balance local details and global context, HoloPart incorporates two key components: (1) a *local attention* design that focuses on capturing the fine-grained geometric details of the input part, and (2) a *shape context-aware attention* mechanism that effectively injects both local and global information to the diffusion model.

To facilitate future research, we propose evaluation benchmarks on the ABO Collins et al. (2022) and PartObjaverse-Tiny Yang et al. (2024) datasets. Extensive experiments demonstrate that HoloPart significantly outperforms existing shape completion approaches. Furthermore, by chaining HoloPart with off-the-shelf 3D part segmentation, we achieve superior results on the full 3D part amodal segmentation task.

In summary, we make the following contributions:

- We formally introduce the task of 3D part amodal segmentation, which separates a 3D shape into multiple semantic parts with complete geometry. This is a critical yet unexplored problem in 3D shape understanding, and provide two new benchmarks (based on ABO and PartObjaverse-Tiny) to facilitate research in this area.

- We propose HoloPart, a novel diffusion-based model for 3D part shape completion. HoloPart features a dual attention mechanism (local attention for fine-grained details and context-aware attention for overall consistency) and leverages a learned 3D generative prior to overcome limitations imposed by scarce training data.

- We demonstrate that HoloPart significantly outperforms existing shape completion methods on the challenging part completion subtask and achieves superior results when integrated with existing segmentation techniques for the full 3D part amodal segmentation task, showcasing its practical applicability and potential for various downstream applications.

## 2 RELATED WORK

**3D Part Segmentation.** 3D Part Segmentation seeks to decompose 3D objects into meaningful, semantic parts, a long-standing challenge in 3D computer vision. Earlier studies Qi et al. (2017a;b); Li

et al. (2018); Zhao et al. (2021); Qian et al. (2022) largely focused on developing network architectures optimized to learn rich 3D representations. These methods generally rely on fully supervised training, which requires extensive, labor-intensive 3D part annotations. Constrained by the limited scale and diversity of available 3D part datasets Mo et al. (2019); Chang et al. (2015), these approaches often face challenges in open-world scenarios. To enable open-world 3D part segmentation, recent methods Liu et al. (2023a); Umam et al. (2023); Kim & Sung (2024); Zhong et al. (2024); Abdelreheem et al. (2023); Tang et al. (2024); Thai et al. (2024); Xue et al. (2023); Yang et al. (2024); Liu et al. (2024) leverage 2D foundation models such as SAM Kirillov et al. (2023), GLIP Li et al. (2022a) and CLIP Radford et al. (2021). These approaches first segment 2D renderings of 3D objects and then develop methods to project these 2D masks onto 3D surfaces. However, due to occlusions, these methods can only segment the visible surface areas of 3D objects, resulting in incomplete segmentations that are challenging to directly apply in downstream tasks. In this work, we advance 3D part segmentation by introducing 3D part amodal segmentation, enabling the completion of segmented parts beyond visible surfaces.

**3D Shape Completion.** 3D shape completion is a post-processing step that restores missing regions, primarily focusing on whole shape reconstruction. Traditional methods like Laplacian hole filling Nealen et al. (2006) and Poisson surface reconstruction Kazhdan et al. (2006) address small gaps and geometric primitives. With the growth of 3D data, retrieval-based methods Sung et al. (2015) have been developed to find and retrieve shapes that best match incomplete inputs from a predefined dataset. The rise of generative models such as GANs Goodfellow et al. (2020), Autoencoders Kingma (2013), and Diffusion models Ho et al. (2020) has led to methods like DiffComplete Chu et al. (2024) and SC-Diff Galvis et al. (2024), which generate diverse and plausible 3D shapes from partial inputs. These models offer flexibility and creative freedom in shape completion.

**3D Shape Diffusion.** Various strategies have been proposed to address the challenges associated with directly training a 3D diffusion model for shape generation, primarily due to the lack of a straightforward 3D representation suitable for diffusion. Several studies Dai et al. (2017); Zhang et al. (2023); Zhao et al. (2024); Zhang et al. (2024) leverage Variational Autoencoders (VAEs) to encode 3D shapes into a latent space, enabling a diffusion model to operate on this latent representation for 3D shape generation. For instance, Shap-E Dai et al. (2017) encodes a point cloud and an image of a 3D shape into an implicit latent space using a transformer-based VAE, enabling subsequent reconstruction as a Neural Radiance Field (NeRF). 3DShape2VecSet Zhang et al. (2023) employs cross-attention mechanisms to encode 3D shapes into latent representations that can be decoded through neural networks. Michelangelo Zhao et al. (2024) further aligns the 3D shape latent space with the CLIP Radford et al. (2021) feature space, enhancing the correspondence between shapes, text, and images. CLAY Zhang et al. (2024) trains a large-scale 3D diffusion model on an extensive dataset, implementing a hierarchical training approach that achieves remarkable results.

## 3  3D Part Amodal Segmentation

We formally introduce the task of *3D part amodal segmentation*. Given a 3D shape $m$, the goal is to decompose $m$ into a set of complete semantic parts, denoted as $\{p_1, p_2, \ldots, p_n\}$, where each $p_i$ represents a geometrically and semantically meaningful region of the shape, *including any occluded portions*. This is in contrast to standard 3D part segmentation, which only identifies visible surface patches. The completed parts should adhere to the following constraints:

1. **Completeness:** Each $p_i$ should represent the entire geometry of the part, even if portions are occluded in the input shape $m$.

2. **Geometric Consistency:** The geometry of each $p_i$ should be plausible and consistent with the visible portions of the part and the overall shape $m$.

3. **Semantic Consistency:** Each $p_i$ should correspond to a semantically meaningful part (e.g., a wheel, a handle).

As discussed in the Introduction, this task presents significant challenges, including inferring occluded geometry, maintaining global shape consistency, and generalizing across diverse shapes and parts, all with limited training data. To address these challenges, we propose a two-stage approach:

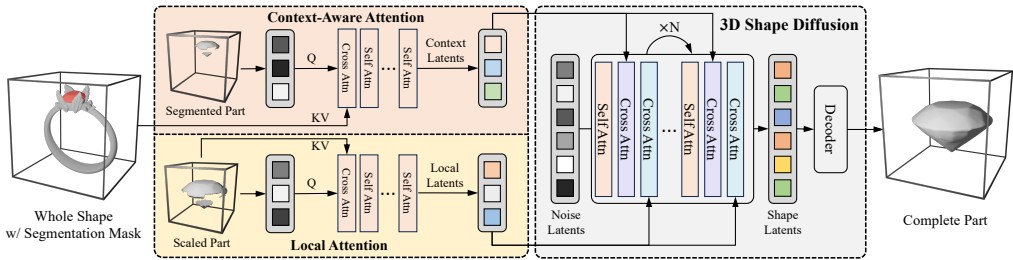

Figure 2: An overview of the **HoloPart** model design. Given a whole 3D shape and a corresponding surface segmentation mask, HoloPart encodes these inputs into latent tokens, using context-aware attention to capture global shape context and local attention to capture local part detailed features and position mapping. These tokens are used as conditions and injected into the part diffusion model via cross-attention respectively. During training, noise is added to complete 3D parts, and the model learns to denoise them and recover the original complete part.

1. **Part Segmentation:** We first obtain an initial part segmentation of the input shape $m$. This provides us with a set of surface patches, each corresponding to a (potentially occluded) semantic segments $\{s_1, s_2, \ldots, s_n\}$. For this stage, we leverage SAMPart3D Yang et al. (2024), although our framework is compatible with other 3D part segmentation techniques.

2. **Part Completion:** This is the core technical contribution of our work. Given an incomplete part segment $s_i$, our goal is to generate the corresponding complete part $p_i$. This requires inferring the missing geometry of the occluded regions while maintaining geometric and semantic consistency. We address this challenge with our **HoloPart** model, described in the following sections.

The remainder of this section details our approach, beginning with the object-level pretraining used to establish a strong 3D generative prior (section 3.1), followed by the key designs of the HoloPart model (section 3.2), and finally the data curation process (section 3.3). The overall pipeline of **HoloPart** is shown in figure 2.

## 3.1 OBJECT-LEVEL PRETRAINING

Due to the scarcity of 3D data with complete part annotations, we first pretrain a 3D generative model on a large-scale dataset of whole 3D shapes. This pretraining allows us to learn a generalizable representation of the 3D shape and capture semantic correspondences between different parts, which is crucial for the subsequent part completion stage.

**Variational Autoencoder (VAE).** We adopt the VAE module design as described in 3DShape2VecSet Zhang et al. (2023) and CLAY Zhang et al. (2024). This design embeds the input point cloud $\mathbf{X} \in \mathbb{R}^{N \times 3}$ sampled from a complete mesh, into a set of latent vectors using a learnable embedding function combined with a cross-attention encoding module:

$$z = \mathcal{E}(\mathbf{X}) = \text{CrossAttn}(\text{PosEmb}(\mathbf{X}_0), \text{PosEmb}(\mathbf{X})), \tag{1}$$

where $\mathbf{X}_0$ represents subsampled point cloud from $\mathbf{X}$ via furthest point sampling, i.e. $\mathbf{X}_0 = \text{FPS}(\mathbf{X}) \in \mathbb{R}^{M \times 3}$. The VAE's decoder, composed of several self-attention layers and a cross-attention layer, processes these latent codes along with a list of query points $q$ in 3D space, to produce the occupancy logits of these positions:

$$\mathcal{D}(z, q) = \text{CrossAttn}(\text{PosEmb}(q), \text{SelfAttn}(z)). \tag{2}$$

**3D Shape Diffusion.** Our diffusion denoising network $v_\theta$ is built upon a series of diffusion transformer (DiT) blocks Peebles & Xie (2023); Zhao et al. (2024); Wu et al. (2024); Zhang et al. (2024); Li et al. (2024). In line with the approach of Rectified Flows (RFs) Liu et al. (2022); Lipman et al. (2022); Albergo & Vanden-Eijnden (2022), our diffusion model is trained in a compressed latent space to map samples from the gaussian distribution $\epsilon \sim \mathcal{N}(0, I)$ to the distribution of 3D shapes. The forward process is defined using a linear interpolation between the original shape and noise, represented as:

$$z_t = (1 - t)z_0 + t\epsilon, \tag{3}$$

where $0 \leq t < 1000$ is the diffusion timestep, $z_0$ represents the original 3D shape, and $z_t$ is progressively noised version of the 3D shape at time $t$. Our goal is to solve the following flow matching objective:

$$\mathbb{E}_{z \in \mathcal{E}(X), t, \epsilon \sim \mathcal{N}(0, I)} \left[ \| v_\theta(z_t, t, g) - (\epsilon - z_0) \|_2^2 \right], \tag{4}$$

where $g$ is the image conditioning feature Wu et al. (2024) derived from the rendering of 3D shape during the pretraining stage.

## 3.2 Context-aware Part Completion

Given a pair consisting of a whole mesh $x$ and a part segment mask $s_i$ on the surface from 3D segmentation models as a prompt, we aim to leverage the learned understanding of 3D shape priors to generate a complete and plausible 3D geometry $p_i$. To preserve local details and capture global context, we incorporate two key mechanisms into our pretrained model: *local attention and shape context-aware attention*. The incomplete part first performs cross-attention with the global shape to learn the contextual shape for completion. Next, the incomplete part is normalized to $[-1, 1]$ and undergoes cross-attention with subsampled points, enabling the model to learn both local details and the new position. Specifically, the context-aware attention and local attention can be expressed as:

$$\begin{aligned} c_o &= \mathcal{C}(\mathbf{S_0}, \mathbf{X}) \\ &= \mathrm{CrossAttn}(\mathrm{PosEmb}(\mathbf{S_0}), \mathrm{PosEmb}(\mathbf{X} \# \# \mathbf{M})), \end{aligned} \tag{5}$$

$$c_l = \mathcal{C}(\mathbf{S_0}, \mathbf{S}) = \mathrm{CrossAttn}(\mathrm{PosEmb}(\mathbf{S_0}), \mathrm{PosEmb}(\mathbf{S})), \tag{6}$$

where $\mathbf{S}$ represents the sampled point cloud on the surface of the incomplete part mesh, and $\mathbf{S_0}$ denotes the subsampled point cloud from $\mathbf{S}$ via furthest point sampling. $\mathbf{X}$ represents the sampled point cloud on the overall shape. Here, $\mathbf{M}$ is a binary mask used to highlight the segmented area on the entire mesh, and ## represents concatenation.

We further finetune the shape diffusion model into a part diffusion model by incorporating our designed local and context-aware attention. The part diffusion model is trained in a compressed latent space to transform noise $\epsilon \sim \mathcal{N}(0, I)$ into the distribution of 3D part shapes. The objective function for part latent diffusion is defined as follows:

$$\mathbb{E}_{z \in \mathcal{E}(K), t, \epsilon \sim \mathcal{N}(0, I)} \left[ \| v_\theta(z_t, t, c_o, c_l) - (\epsilon - z_0) \|_2^2 \right], \tag{7}$$

where $K$ represents the sampled point cloud from the complete part meshes. Following Zhao et al. (2024), we apply classifier-free guidance (CFG) by randomly setting the conditional information to a zero vector randomly. Once the denoising network $v_\theta$ is trained, the function $f$ can generate $\hat{m}_p$ by iterative denoising. The resulting latent embedding is then decoded into 3D space occupancy and the mesh is extracted from the part region using the marching cubes Lorensen & Cline (1998).

## 3.3 Data Curation

We process data from two 3D datasets: ABO Collins et al. (2022) and Objaverse Deitke et al. (2023). For the ABO dataset, which contains part ground truths, we directly use this information to generate whole-part pair data. In contrast, filtering valid part data from Objaverse is challenging due to the absence of part annotations, and the abundance of scanned objects and low-quality models. To address this, we first filter out all scanned objects and select 180k high-quality 3D shapes from the original 800,000 available models. We then develop a set of filtering rules to extract 3D objects with a reasonable part-wise semantic distribution from 3D asset datasets, including Mesh Count Restriction, Connected Component Analysis and Volume Distribution Optimization. Further details are provided in the supplementary.

To train the conditional part diffusion model $f$, we develop a data creation pipeline to generate whole-part pair datasets. First, all component parts are merged to form the complete 3D mesh. Next, several rays are sampled from different angles to determine the visibility of each face, and any invisible faces are removed. To handle non-watertight meshes, we compute the Unsigned Distance Field (UDF) of the 3D mesh and then obtain the processed whole 3D mesh using the marching cubes algorithm. We apply a similar process to each individual 3D part to generate the corresponding complete 3D part mesh. Finally, we assign part labels to each face of the whole mesh by finding the nearest part face, which provides surface segment masks $\{s_i\}$.

# 4 EXPERIMENTS

| | | P/C | D/C | F/V | Ours w/o C-a | Ours w C-a |
|---|---|---|---|---|---|---|
| Chamfer ↓ | bed | 0.093 | 0.061 | 0.023 | 0.032 | **0.020** |
| | table | 0.081 | 0.068 | 0.030 | 0.042 | **0.018** |
| | lamp | 0.170 | 0.084 | 0.044 | 0.036 | **0.031** |
| | chair | 0.121 | 0.107 | 0.045 | 0.035 | **0.030** |
| | mean (instance) | 0.122 | 0.087 | 0.037 | 0.036 | **0.026** |
| | mean (category) | 0.116 | 0.080 | 0.035 | 0.036 | **0.025** |
| IoU ↑ | bed | 0.148 | 0.266 | 0.695 | 0.792 | **0.833** |
| | table | 0.180 | 0.248 | 0.652 | 0.791 | **0.838** |
| | lamp | 0.155 | 0.238 | 0.479 | 0.677 | **0.697** |
| | chair | 0.156 | 0.214 | 0.490 | 0.695 | **0.718** |
| | mean (instance) | 0.159 | 0.235 | 0.565 | 0.733 | **0.764** |
| | mean (category) | 0.160 | 0.241 | 0.580 | 0.739 | **0.771** |
| F1-Score ↑ | bed | 0.244 | 0.412 | 0.802 | 0.864 | **0.896** |
| | table | 0.291 | 0.390 | 0.758 | 0.844 | **0.890** |
| | lamp | 0.244 | 0.374 | 0.610 | 0.769 | **0.789** |
| | chair | 0.262 | 0.342 | 0.631 | 0.800 | **0.817** |
| | mean (instance) | 0.259 | 0.371 | 0.689 | 0.816 | **0.843** |
| | mean (category) | 0.260 | 0.380 | 0.700 | 0.819 | **0.848** |
| Success ↑ | mean (instance) | 0.822 | 0.824 | 0.976 | 0.987 | **0.994** |

Table 1: 3D part amodal segmentation results of PatchComplete, DiffComplete, Finetune-VAE, Ours (w/o Context-attention), Ours (with Context-attention), on ABO, reported in Chamfer Distance, IoU, F-Score and Success Rate.

| | | P/C | D/C | S/F | Ours |
|---|---|---|---|---|---|
| Chamfer ↓ | car | 0.289 | 0.153 | 0.264 | **0.090** |
| | airplane | 0.267 | 0.141 | 0.241 | **0.087** |
| | faucet | 0.258 | 0.125 | 0.232 | **0.076** |
| | bed | 0.295 | 0.162 | 0.282 | **0.097** |
| | mean (instance) | 0.278 | 0.146 | 0.255 | **0.088** |
| | mean (category) | 0.277 | 0.145 | 0.255 | **0.087** |
| IoU ↑ | car | 0.247 | 0.382 | 0.323 | **0.545** |
| | airplane | 0.231 | 0.405 | 0.230 | **0.572** |
| | faucet | 0.291 | 0.442 | 0.185 | **0.601** |
| | bed | 0.215 | 0.368 | 0.254 | **0.531** |
| | mean (instance) | 0.245 | 0.401 | 0.246 | **0.558** |
| | mean (category) | 0.246 | 0.399 | 0.248 | **0.562** |
| F1-Score ↑ | car | 0.314 | 0.485 | 0.406 | **0.635** |
| | airplane | 0.291 | 0.508 | 0.299 | **0.652** |
| | faucet | 0.365 | 0.529 | 0.277 | **0.673** |
| | bed | 0.282 | 0.416 | 0.313 | **0.614** |
| | mean (instance) | 0.312 | 0.485 | 0.321 | **0.641** |
| | mean (category) | 0.313 | 0.484 | 0.323 | **0.644** |
| Success ↑ | mean (instance) | 0.835 | 0.935 | 0.884 | **0.995** |

Table 2: 3D part amodal segmentation results of PatchComplete, DiffComplete, SD-Fusion and Ours on 3DCoMPaT++ with 2.5D mask input, reported in Chamfer Distance, IoU, F-Score and Success Rate.

## 4.1 EXPERIMENTAL SETUP

**Datasets and Benchmarks.** We propose two benchmarks based on two 3D shape datasets: ABO Collins et al. (2022) and PartObjaverse-Tiny Yang et al. (2024), to evaluate the 3D amodal segmentation task. The ABO dataset contains high-quality 3D models of real-world household objects, covering four categories: bed, table, lamp, and chair, all with detailed part annotations. For training, we use 20,000 parts, and for evaluation, we use 1,000 parts (60 shapes). Objaverse Deitke et al. (2023) is a large-scale 3D dataset comprising over 800,000 3D shapes. PartObjaverse-Tiny is a curated subset of Objaverse, consisting of 3,000 parts (200 shapes) with fine-grained part annotations. Using our data-processing pipeline, we construct two evaluation datasets. Specifically, we project the ground-truth masks onto the surfaces of the processed monolithic meshes to serve as inputs, and use the complete part annotations as the outputs (targets) for evaluation. We further evaluate the 3D amodal segmentation task by replacing the ground-truth masks with masks produced by SAMPart3D. We further investigate the generalization capabilities of our model. Specifically, we demonstrate that our method can be integrated with arbitrary 3D part surface segmentation models to process "holistic shells" (generated or scanned), yielding complete and consistent parts. Moreover, we extend our exploration to the task of 2.5D part completion on the 3DCoMPaT++ Slim et al. (2025); Li et al. (2022b) dataset.

**Baselines.** We compare our methods against state-of-the-art shape completion models, PatchComplete Rao et al. (2022), DiffComplete Chu et al. (2024) and SDFusion Cheng et al. (2023) using our proposed benchmarks. We train all baselines on our processed ABO and Objaverse datasets using the official implementations. To adapt to the data requirements of these models, we generated voxel grids with SDF values from our processed meshes. Additionally, our VAE model also uses 3D encoder-decoder architectures for 3D shape compression and reconstruction. Thus, we directly fine-tune the VAE on our parts dataset for part completion, serving as a baseline method.

**Metrics.** To evaluate the quality of predicted part shape geometry, we use three metrics: $\mathcal{L}_1$ Chamfer Distance (CD) Intersection over Union (IoU), and F-Score, comparing the predicted and ground truth part shapes. We sample 500k points on both the predicted and the group truth part meshes to capture detailed geometry information, used for the CD calculation. To compute IoU and F-Score, we generate voxel grids of size $64^3$ with occupancy values based on the sampled points. Since the baseline methods are sometimes unable to reconstruct effective meshes, we calculate CD, IoU, and F-Score only for the successfully reconstructed meshes. Additionally, we report the reconstruction success ratio to quantify the reliability of each method.

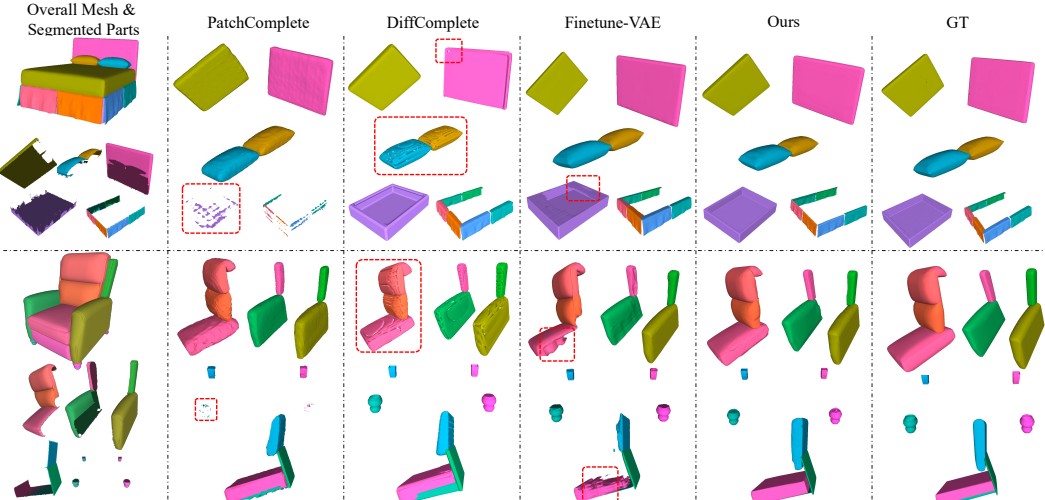

Figure 3: Qualitative comparison with PatchComplete, DiffComplete and Finetune-VAE on the ABO dataset.

| | Method | Overall | Human | Animals | Daily | Buildings | Transports | Plants | Food | Electronics |
|---|---|---|---|---|---|---|---|---|---|---|
| Chamfer ↓ | PatchComplete | 0.144 | 0.150 | 0.165 | 0.141 | 0.173 | 0.147 | 0.110 | 0.118 | 0.147 |
| | DiffComplete | 0.133 | 0.130 | 0.144 | 0.127 | 0.145 | 0.136 | 0.129 | 0.128 | 0.125 |
| | SDFusion | 0.137 | 0.135 | 0.162 | 0.146 | 0.162 | 0.144 | 0.104 | 0.105 | 0.134 |
| | Finetune-VAE | 0.064 | 0.064 | 0.067 | 0.075 | 0.064 | 0.076 | 0.049 | 0.041 | 0.073 |
| | Ours w/o Local | 0.057 | 0.061 | 0.083 | 0.051 | 0.047 | 0.075 | 0.045 | 0.037 | 0.057 |
| | Ours w/o Context | 0.055 | 0.059 | 0.076 | 0.044 | 0.047 | 0.053 | 0.042 | 0.039 | 0.056 |
| | **Ours** | **0.034** | **0.034** | **0.042** | **0.032** | **0.032** | **0.037** | **0.029** | **0.029** | **0.041** |
| IoU ↑ | PatchComplete | 0.137 | 0.129 | 0.147 | 0.132 | 0.116 | 0.129 | 0.152 | 0.156 | 0.138 |
| | DiffComplete | 0.142 | 0.149 | 0.139 | 0.142 | 0.124 | 0.139 | 0.153 | 0.134 | 0.157 |
| | SDFusion | 0.235 | 0.214 | 0.237 | 0.229 | 0.202 | 0.198 | 0.265 | 0.294 | 0.242 |
| | Finetune-VAE | 0.502 | 0.460 | 0.464 | 0.503 | 0.513 | 0.468 | 0.536 | 0.583 | 0.490 |
| | Ours w/o Local | 0.618 | 0.582 | 0.574 | 0.618 | 0.634 | 0.591 | 0.673 | 0.677 | 0.594 |
| | Ours w/o Context | 0.553 | 0.535 | 0.518 | 0.579 | 0.593 | 0.553 | 0.590 | 0.609 | 0.538 |
| | **Ours** | **0.688** | **0.675** | **0.667** | **0.699** | **0.714** | **0.687** | **0.709** | **0.710** | **0.648** |
| F1-Score ↑ | PatchComplete | 0.232 | 0.221 | 0.246 | 0.224 | 0.197 | 0.220 | 0.254 | 0.261 | 0.233 |
| | DiffComplete | 0.239 | 0.250 | 0.235 | 0.238 | 0.212 | 0.234 | 0.254 | 0.225 | 0.262 |
| | SDFusion | 0.365 | 0.340 | 0.368 | 0.357 | 0.318 | 0.316 | 0.403 | 0.442 | 0.374 |
| | Finetune-VAE | 0.638 | 0.600 | 0.613 | 0.638 | 0.646 | 0.596 | 0.672 | 0.718 | 0.623 |
| | Ours w/o Local | 0.741 | 0.715 | 0.706 | 0.743 | 0.750 | 0.713 | 0.786 | 0.796 | 0.719 |
| | Ours w/o Context | 0.691 | 0.679 | 0.663 | 0.716 | 0.722 | 0.688 | 0.727 | 0.743 | 0.676 |
| | **Ours** | **0.801** | **0.794** | **0.788** | **0.809** | **0.818** | **0.798** | **0.817** | **0.820** | **0.767** |
| Success ↑ | PatchComplete | 0.938 | 0.989 | 0.976 | 0.954 | 0.843 | 0.932 | 0.959 | 0.935 | 0.947 |
| | DiffComplete | 0.942 | 0.992 | 0.980 | 0.958 | 0.851 | 0.936 | 0.959 | 0.935 | 0.950 |
| | Finetune-VAE | 0.997 | **1.000** | **0.996** | 0.996 | **0.997** | 0.996 | **1.000** | **1.000** | 0.994 |
| | Ours w/o Context | **0.998** | 0.997 | **0.996** | **1.000** | **0.997** | **0.994** | **1.000** | **1.000** | **0.997** |

Table 3: 3D part amodal segmentation results on PartObjaverse-Tiny, reported in Chamfer Distance, IoU, F-Score and Success Rate.

## 4.2 MAIN RESULTS

**ABO.** We compare our method with PatchComplete Rao et al. (2022), DiffComplete Chu et al. (2024) and our fintuned VAE on the ABO dataset. Quantitative results are presented in table 1, with qualitative comparisons illustrated in figure 3. When dealing with parts containing large missing areas, PartComplete struggles to generate a plausible shape. PatchComplete and DiffComplete often fail to reconstruct small or thin structures, such as the bed sheets or the connections of the lamp in figure 3. Although the finetuned VAE can reconstruct parts that have substantial visible areas, it performs poorly when completing regions with little visibility, such as the bedstead or the interior of the chair, as shown in figure 3. In contrast, our method consistently generates high-quality, coherent parts and significantly outperforms the other approaches in both quantitative and qualitative evaluations.

**PartObjaverse-Tiny.** We also compare our method with PatchComplete, DiffComplete, and our finetuned VAE on the PartObjaverse-Tiny dataset. The shapes in the PartObjaverse-Tiny dataset are more complex and diverse, making part completion more challenging. We calculate the Cham-

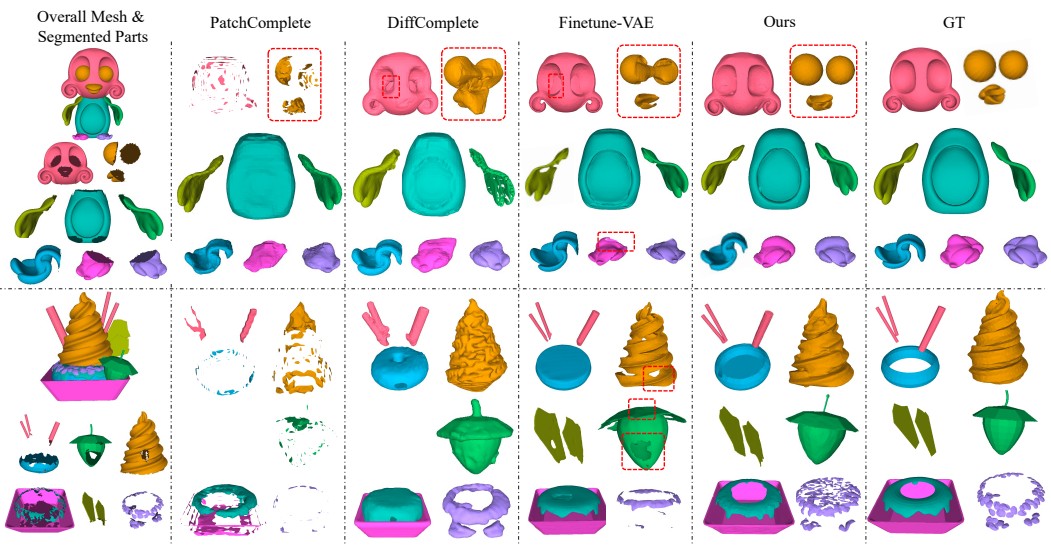

Figure 4: Qualitative comparison on the PartObjaverse-Tiny dataset.

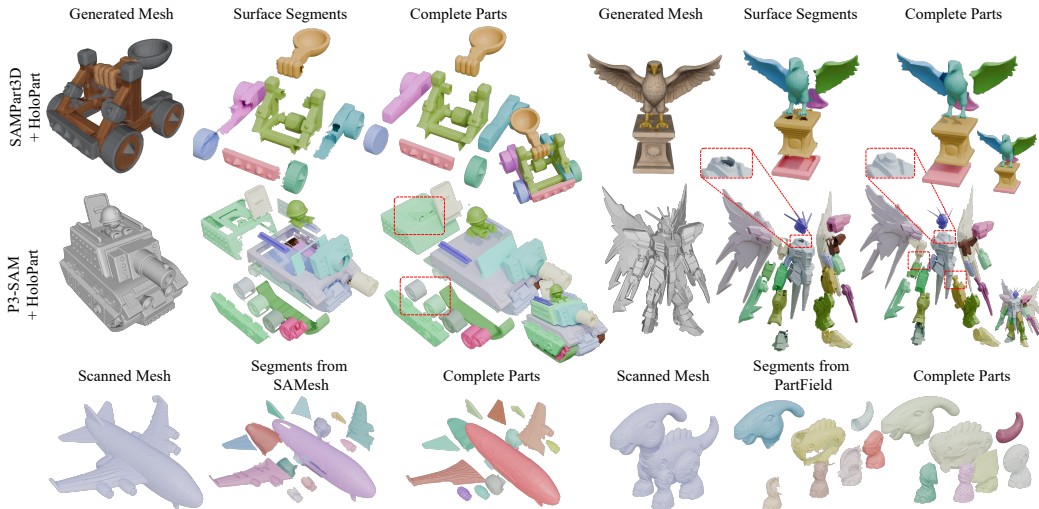

Figure 5: 3D part amodal segmentation on **generated objects and scanned objects from OmniObject3D**. Our method seamlessly integrates with arbitrary zero-shot 3D part segmentation models. We can generate even precise **joint structures**, such as the mortise-and-tenon joints at the robot's connections shown in the figure.

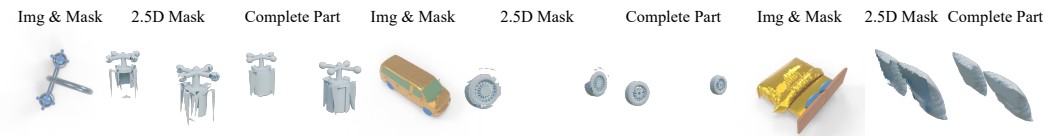

Figure 6: 3D part amodal segmentation on 3DCoMPaT++ Slim et al. (2025) with 2.5D mask input.

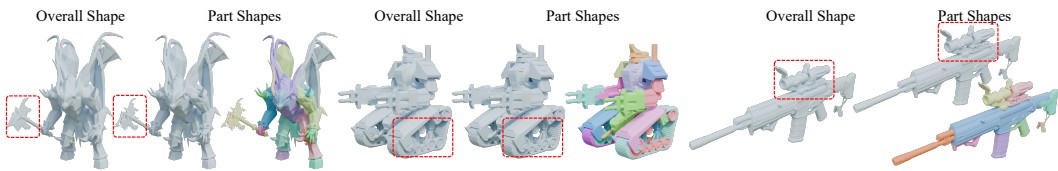

Figure 7: Geometry Super-resolution. By representing a part with the same number of tokens as the overall object, we can achieve geometry super-resolution.

| | Method | Overall | Human | Animals | Daily | Buildings | Transports | Plants | Food | Electronics |
|---|---|---|---|---|---|---|---|---|---|---|
| Chamfer ↓ | SDFusion | 0.264 | 0.241 | 0.232 | 0.282 | 0.365 | 0.323 | 0.230 | 0.185 | 0.254 |
| | PatchComplete | 0.289 | 0.267 | 0.258 | 0.295 | 0.382 | 0.314 | 0.247 | 0.231 | 0.291 |
| | DiffComplete | 0.231 | 0.197 | 0.193 | 0.252 | 0.307 | 0.264 | 0.206 | 0.198 | 0.235 |
| | Finetune-VAE | 0.178 | 0.138 | 0.114 | 0.202 | 0.279 | 0.213 | 0.140 | 0.141 | 0.198 |
| | Ours | **0.134** | **0.094** | **0.086** | **0.155** | **0.210** | **0.144** | **0.109** | **0.110** | **0.162** |
| IoU ↑ | SDFusion | 0.169 | 0.159 | 0.191 | 0.161 | 0.124 | 0.117 | 0.201 | 0.234 | 0.168 |
| | PatchComplete | 0.086 | 0.079 | 0.097 | 0.079 | 0.076 | 0.076 | 0.105 | 0.091 | 0.084 |
| | DiffComplete | 0.102 | 0.115 | 0.121 | 0.093 | 0.073 | 0.087 | 0.122 | 0.109 | 0.098 |
| | Finetune-VAE | 0.347 | 0.370 | 0.406 | 0.313 | 0.299 | 0.277 | 0.412 | 0.381 | 0.320 |
| | Ours | **0.455** | **0.508** | **0.513** | **0.415** | **0.360** | **0.379** | **0.522** | **0.529** | **0.416** |
| F1-Score ↑ | SDFuison | 0.273 | 0.263 | 0.306 | 0.260 | 0.208 | 0.198 | 0.316 | 0.364 | 0.271 |
| | PatchComplete | 0.149 | 0.139 | 0.168 | 0.138 | 0.133 | 0.134 | 0.179 | 0.157 | 0.147 |
| | DiffComplete | 0.177 | 0.198 | 0.206 | 0.162 | 0.129 | 0.153 | 0.206 | 0.189 | 0.170 |
| | Finetune-VAE | 0.473 | 0.507 | 0.543 | 0.433 | 0.417 | 0.395 | 0.540 | 0.513 | 0.439 |
| | Ours | **0.570** | **0.626** | **0.628** | **0.529** | **0.477** | **0.497** | **0.627** | **0.645** | **0.533** |
| Success ↑ | PatchComplete | 0.978 | 0.992 | 0.998 | 0.992 | 0.957 | 0.975 | 0.988 | **1.000** | 0.966 |
| | DiffComplete | 0.942 | 0.992 | 0.980 | 0.958 | 0.851 | 0.936 | 0.959 | 0.935 | 0.950 |
| | Finetune-VAE | 0.997 | **1.000** | 0.992 | 0.992 | 0.995 | **1.000** | 0.994 | **1.000** | 0.997 |
| | Ours | **1.000** | **1.000** | **1.000** | **1.000** | **1.000** | **1.000** | **1.000** | **1.000** | **1.000** |

Table 4: 3D part amodal segmentation results on PartObjaverse-Tiny, using SAMPart3D's segment masks as input, reported in Chamfer Distance, IoU, F-Score and Success Rate.

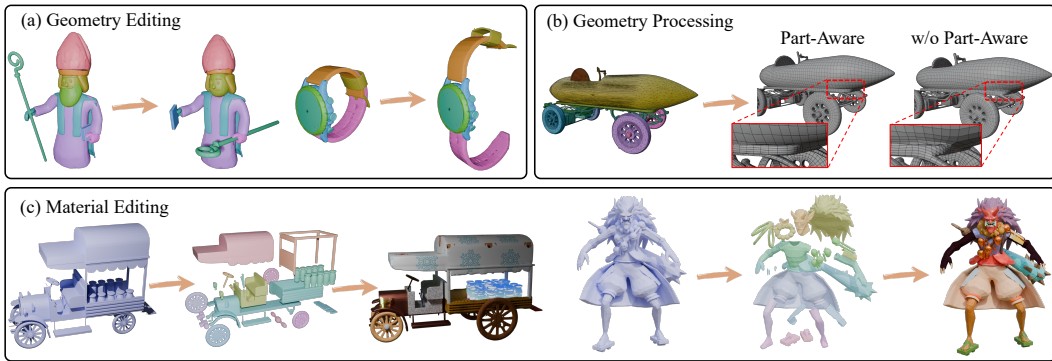

Figure 8: 3D part amodal segmentation is capable of numerous downstream applications, such as Geometry Editing, Geometry Processing, Material Editing and Animation.

fer Distance, IoU, F-Score, and Reconstruction Success rate for each method, with the quantitative comparison shown in table 3. Our method consistently outperforms the others, even on this challenging dataset. As shown in figure 4, our approach effectively completes intricate details, such as the eyeball, strawberry, which the other methods fail to achieve.

**3DCoMPaT++.** By back-projecting a 2D rendered image and its corresponding 2D mask onto the 3D mesh surface, we can acquire the corresponding 2.5D mask information. Based on this, we can complete the geometry of the parts visible in the image. Leveraging the fine-grained annotations and 2D renderings from the 3DCoMPaT++ dataset Slim et al. (2025); Li et al. (2022b), we select four suitable categories—car, airplane, faucet, and bed—for evaluation. In figure 6, we visualize the performance of our method on visible parts, while table 2 presents a quantitative comparison with baseline methods.

**Zero-shot Generalization.** By leveraging pretraining on the large-scale Objaverse dataset and fine-tuning on processed parts data, our model is capable of zero-shot amodal segmentation. To demonstrate the generalization capabilities of our model in a challenging zero-shot setting, we present 3D part amodal sementation results on generated meshes and the scanned dataset OmniObject3D Wu et al. (2023). Current generated and scanned objects typically exist as "holistic shells", lacking complete part geometry. Our method addresses this by seamlessly integrating with arbitrary surface segmentation models to recover a full set of parts, achieving a structural quality that closely resembles artificial models. For instance, figure 5 demonstrates our method applied to generated objects using SAMPart3D Yang et al. (2024) and P3-SAM Ma et al. (2025) as segmentation methods. Notably, our model is capable of generating the internal connecting structures between parts.

Furthermore, figure 5 also presents the results on scanned objects, leveraging SAMesh Tang et al. (2024) and PartField Liu et al. (2025) as segmentation methods.

| Method | Chamfer ↓ | F1-0.1 ↑ | F1-0.05 ↑ |
|--------|-----------|----------|-----------|
| TripoSG | 0.120 | 0.828 | 0.626 |
| Ours | **0.114** | **0.834** | **0.667** |

Table 5: Comparison with TripoSG. By generating complete and coherent parts, our method achieves better performance.

| Guidance Scale | Chamfer ↓ | IoU ↑ | F1-Score ↑ | Success ↑ |
|----------------|-----------|-------|------------|-----------|
| $S = 1.5$ | 0.059 | 0.590 | 0.718 | 0.995 |
| $S = 3.5$ | **0.057** | **0.618** | **0.741** | **0.997** |
| $S = 5$ | 0.058 | 0.614 | 0.738 | 0.996 |
| $S = 7.5$ | 0.089 | 0.514 | 0.641 | **0.997** |

Table 6: Ablation study of different guidance scales.

### 4.3 ABLATION ANALYSIS

**Necessity of Context-Aware Attention.** The context-aware attention is crucial for completing invisible areas of parts and ensuring the consistency of generated components. To demonstrate this, we replace the context-aware attention block with a local-condition block and train the model. The quantitative comparison shown in table 1 and table 3 demonstrates the significance of context-aware attention. The qualitative analysis is provided in the supplementary material.

**Necessity of Local Attention.** Local attention is crucial for maintaining details and mapping positions. We perform an ablation study on the local attention module and present the quantitative comparison in table 3, highlighting the necessity of our local attention design.

**Effect of Guidance Scale.** We find that the guidance scale significantly impacts the quality of our generated shapes. We evaluate four different guidance scales (1.5, 3.5, 5, and 7) on the PartObjaverse-Tiny dataset, with the results presented in table 6. A small guidance scale leads to insufficient control, while an excessively large guidance scale results in the failure of shape reconstruction from latent fields. We find a scale of 3.5 provides the optimal balance.

### 4.4 APPLICATION

Our model is capable of completing high-quality parts across a variety of 3D shapes, thereby enabling numerous downstream applications such as **geometry editing**, **material assignment** and **animation**. We demonstrate the application of geometry editing in Figures 1 and 8 (a), and material assignment in Figures 1 and 8 (c). For example, in the case of the car model, we perform 3D part amodal segmentation, then modify the sizes of the front and rear wheels, increase the number of jars, and expand the car's width in Blender. Afterward, we assign unique textures to each part and enable the wheels and steering wheel to move. The video demo is included in the supplementary material. These operations would be difficult to achieve with traditional 3D part segmentation techniques. Additionally, we showcase an example of a geometry processing application in Figure 8 (b).

Our model also has the potential for **Geometric Super-resolution**. By representing a part with the same number of tokens as the overall object, we can fully preserve and generate the details of the part. A comparison with the overall shape, reconstructed using the same number of tokens by VAE, is shown in figure 7.

## 5 CONCLUSION

This paper introduces 3D part amodal segmentation, a novel task that addresses a key limitation in 3D content generation. We decompose the problem into subtasks, focusing on 3D part shape completion, and propose a diffusion-based approach with local and context-aware attention mechanisms to ensure coherent part completion. We establish evaluation benchmarks on the ABO and PartObjaverse-Tiny datasets, demonstrating that our method significantly outperforms prior shape completion approaches. Our comprehensive evaluations and application demonstrations validate the effectiveness of our approach and establish a foundation for future research in this emerging field.

## 6 Acknowledgements

The research work described in this paper was conducted in the JC STEM Lab of Autonomous Intelligent Systems funded by The Hong Kong Jockey Club Charities Trust.

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

# A APPENDIX

## A.1 THE USE OF LARGE LANGUAGE MODELS (LLMS)

We use Large Language Models (LLMs) strictly for minor language editing—grammar and readability, not for method design or experiments. All technical contributions, including the methodology, equations, and results, are solely the work of the authors.

## A.2 IMPLEMENTATION DETAILS

The VAE consists of 24 transformer blocks, with 8 blocks functioning as the encoder and the remaining 16 as the decoder. The part diffusion model consists of 10 DiT layers with a hidden size of 2048, and the context-aware attention block consists of 8 self-attention blocks. To balance effectiveness with training efficiency, we set the token number for our part diffusion to 512. The latent tokens, encoded by the context-aware attention block, have a dimension of (512, 512), which are integrated into the part diffusion model via cross-attention. We fine-tune the part diffusion model using the ABO Collins et al. (2022) dataset with 4 RTX 4090 GPUs for approximately two days, using the Objaverse Deitke et al. (2023) dataset with 8 A100 GPUs for around four days.

We set the learning rate to 1e-4 for both the pretraining and finetuning stages, using the AdamW optimizer. During training, as illustrated in figure 2, we sample 20,480 points from the overall shape, which serve as the keys and values, while 512 points are sampled from each segmented part to serve as the query. This results in the context latent dimensions being (512, 512). For each point, we use the position embedding concatenated with a normal value as the input feature. After passing through the denoising UNet, we obtain shape latents of dimensions (512, 2048), representing the complete part's shape. Subsequently, we use the 3D spatial points to query these shape latents and employ a local marching cubes algorithm to reconstruct the complete part mesh. The local bounding box is set to be 1.3 times the size of the segmented part's bounding box to ensure complete mesh extraction.

## A.3 DATA CURATION DETAILS

We develop a set of filtering rules to extract 3D objects with a reasonable part-wise semantic distribution from 3D asset datasets. We ultimately retain 16,000 parts (50000 objects) in Objaverse as training data. The specific rules are as follows:

- **Mesh Count Restriction**: We select only 3D objects with a mesh count within a specific range (2 to 15) to avoid objects that are either too simple or too complex (such as scenes or architectural models). The example data filtered out by this rule is shown in figure 11 (a).

- **Connected Component Analysis**: For each object, we render both frontal and side views of all parts and calculate the number of connected components in the 2D images. We then compute the average number of connected components per object, as well as the top three average values. An empirical threshold (85% of the connected component distribution) is used to filter out objects with severe fragmentation or excessive floating parts (floaters). The example data filtered out by this rule is shown in figure 11 (b).

- **Volume Distribution Optimization**: We analyze the volume distribution among different parts and ensure a balanced composition by removing or merging small floating parts and filtering out objects where a single part dominates excessively (e.g., cases where the alpha channel of the rendered image overlaps with the model rendering by up to 90%). The example data filtered out by this rule is shown in figure 11 (c).

## A.4 MORE ABLATION ANALYSIS

**Semantic and Instance Part Completion.** Traditionally, segmentation definitions fall into two categories: semantic segmentation and instance segmentation. Similarly, we process our 3D parts from the ABO dataset according to these two settings. For example, in the semantic part completion setting, we consider all four chair legs as a single part, whereas in the instance part completion setting, they are treated as four separate parts. Our model is capable of handling both settings

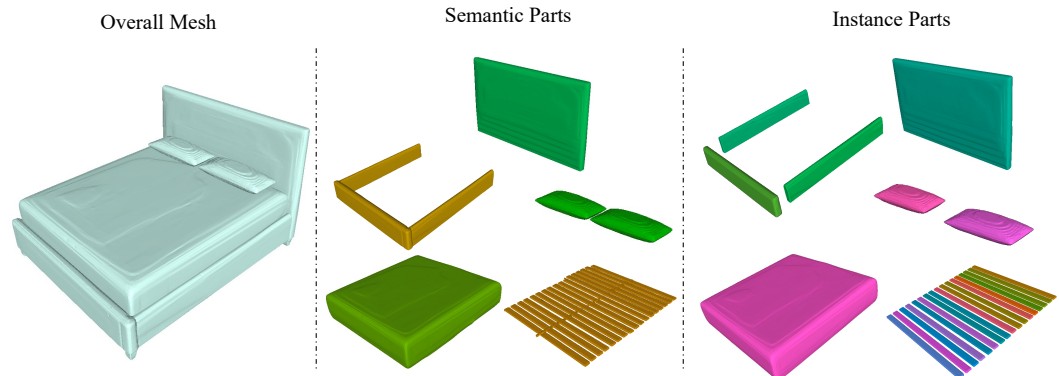

Figure 9: Ablation study of semantic and instance part completion.

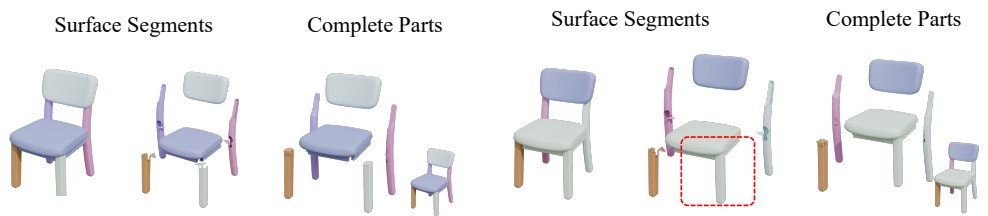

Figure 10: Ablation study on segmentation mask input with different levels of granularity.

effectively. We train on the mixed dataset and present the completion results for a single bed using the same model weight, as shown in figure 9.

**Necessity of Context-Aware Attention.** To emphasize the importance of our proposed context-aware attention block, we provide both quantitative analysis (refer to section 4.3) and qualitative comparisons. As shown in figure 12, the absence of context-aware attention results in a lack of guidance for completing individual parts, leading to inconsistent and lower-quality completion outcomes.

**Qualitative Comparison of Different Guidance Scales.** In section 4.3, we provide a quantitative analysis of various guidance scales. Additionally, We illustrate the qualitative comparison of different guidance scales in figure 13. Our findings indicate that excessively large or small guidance scales can adversely impact the final completion results. Through experimentation, we identify 3.5 as an optimal value for achieving balanced outcomes.

**Learning Rate Setting.** During the fine-tuning stage, we experiment with a weighted learning rate approach, where the parameters of the denoising U-Net are set to 0.1 times that of the context-aware attention block. However, we observe that this approach results in unstable training and negatively impacts the final outcomes. We present the comparison of generated parts with different learning rate training setting in figure 13.

**Ambiguity of Segmentation Mask.** Our model is robust to different levels of segmentation granularity. As shown in figure 10, whether the chair's leg and seat are separated or merged during the segmentation stage does not affect the final quality.

## A.5 MORE RESULTS OF 3D PART AMODAL SEGMENTATION

In figure 15, we showcase additional examples of 3D part amodal segmentation applied to generated meshes from 3D generation models. Initially, we employ SAMPart3D Yang et al. (2024) to segment the generated meshes, resulting in several surface masks. Subsequently, our model completes each segmented part, enabling the reconstruction of a consistent overall mesh by merging the completed parts. For instance, as demonstrated in figure 15, our model effectively completes intricate com-

ponents such as glasses, hats, and headsets from the generated meshes. This capability supports a variety of downstream tasks, including geometry editing, geometry processing, and material editing.

### A.6 MORE RESULTS ON PARTOBJAVERSE-TINY

We present more qualitative results on the PartObjaverse-Tiny dataset in Figures 17 and 16. Our method can effectively complete the details of parts and maintain overall consistency, which other methods cannot achieve.

### A.7 LIMITATIONS AND FUTURE WORKS

The outcome of HoloPart is influenced by the quality of input surface masks. Unreasonable or low-quality masks may lead to incomplete results. Therefore, a better approach moving forward would be to use our method to generate a large number of 3D part-aware shapes, which can then be used to train part-aware generation models.

(a)

(b)

(c)

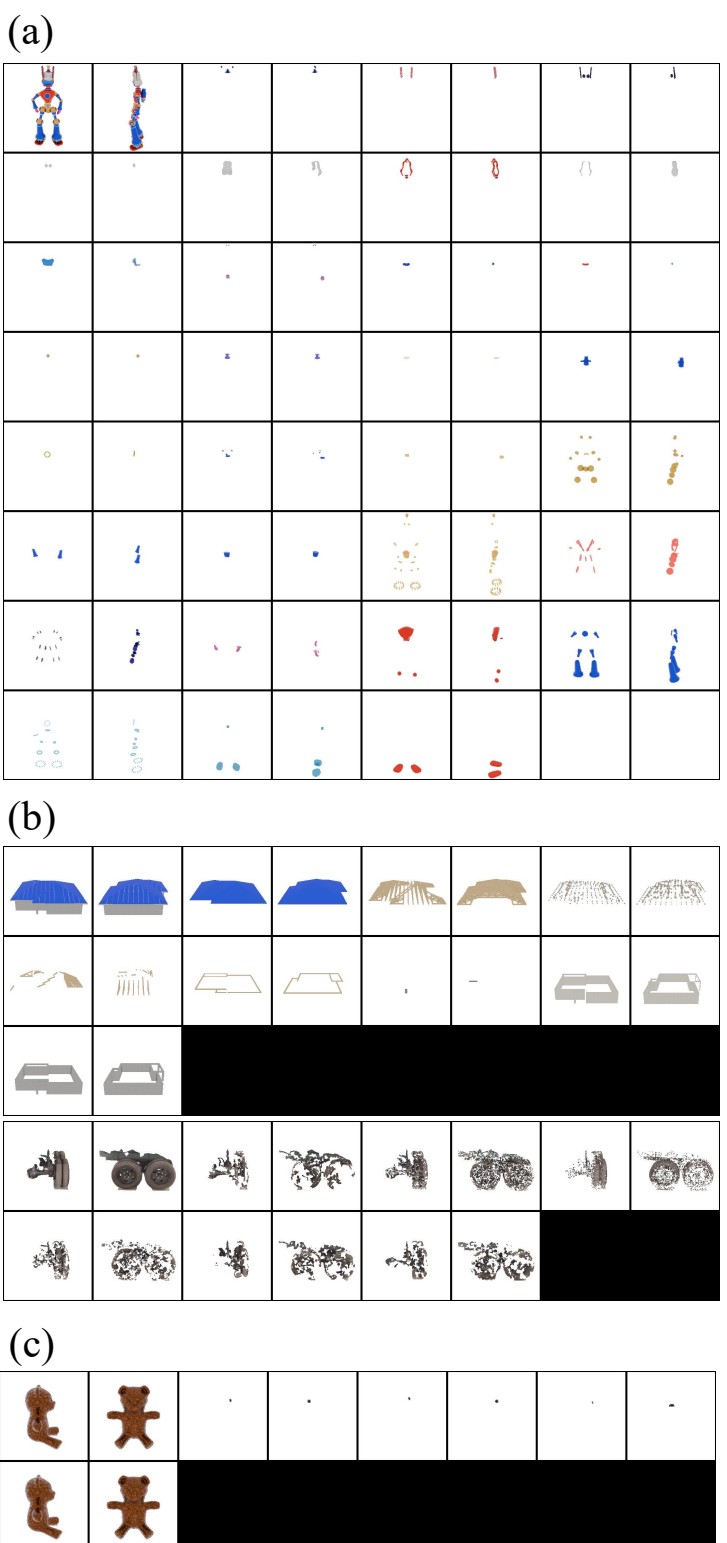

Figure 11: Examples of data filtered out by rules.

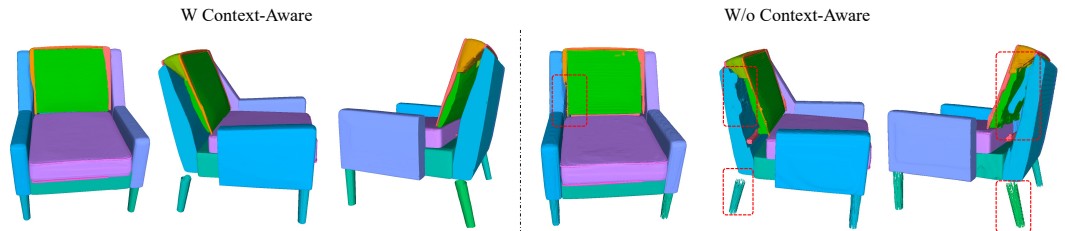

Figure 12: The absence of context-aware attention leads to a lack of guidance for completing individual components, resulting in inconsistent and lower-quality outcomes.

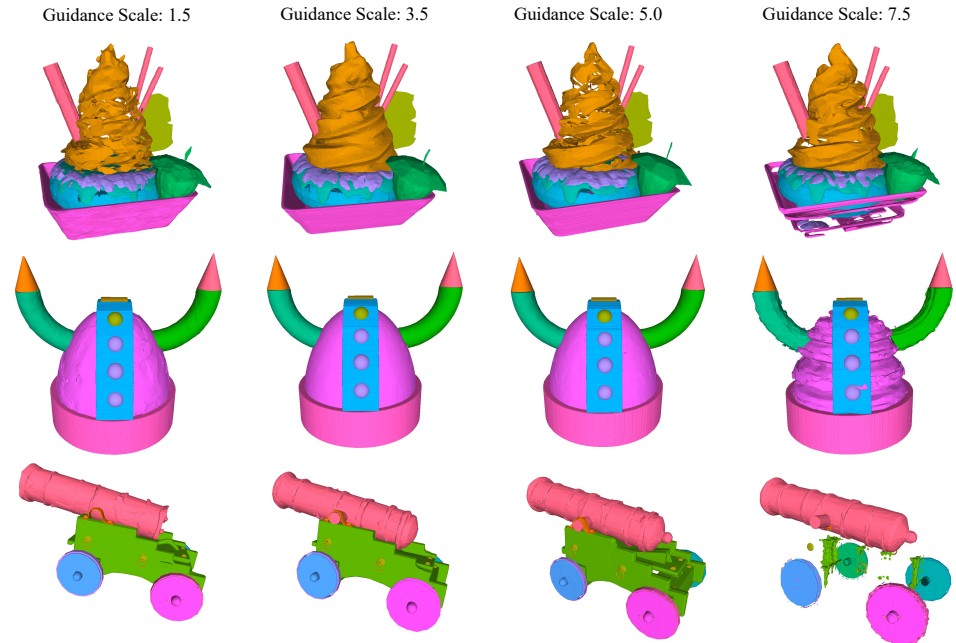

Figure 13: Visualization of generated parts across different guidance scales.

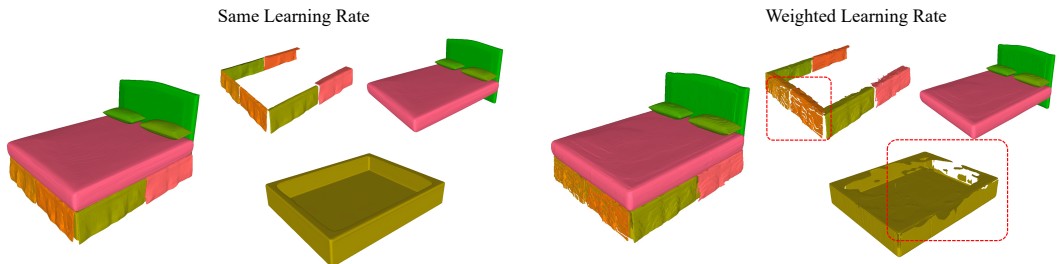

Figure 14: Qualitative comparison of different learning rate settings.

Generated Mesh    Surface Segments    Complete Parts    Merged Parts

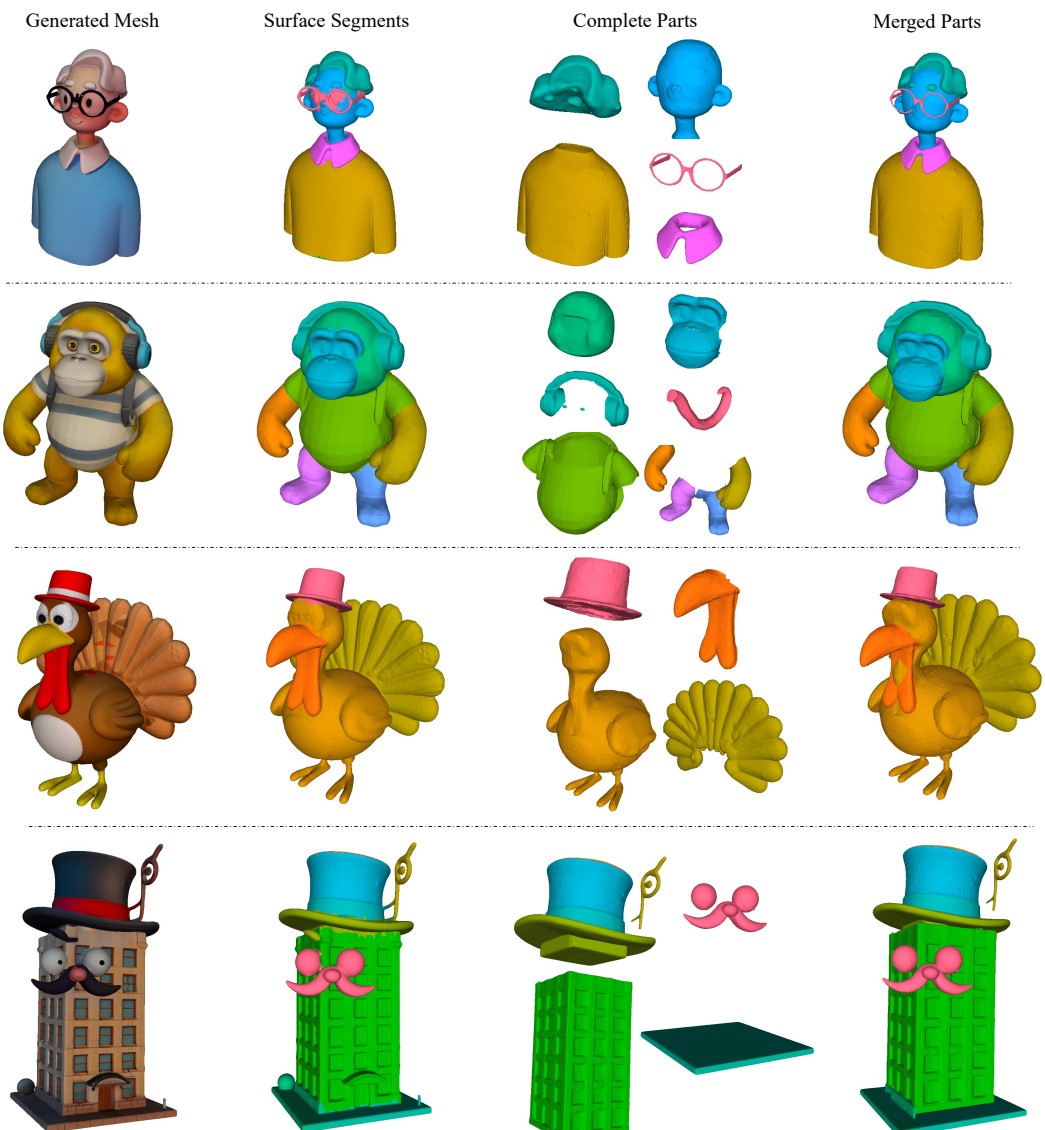

Figure 15: More Results of 3D Part Amodal Segmentation.

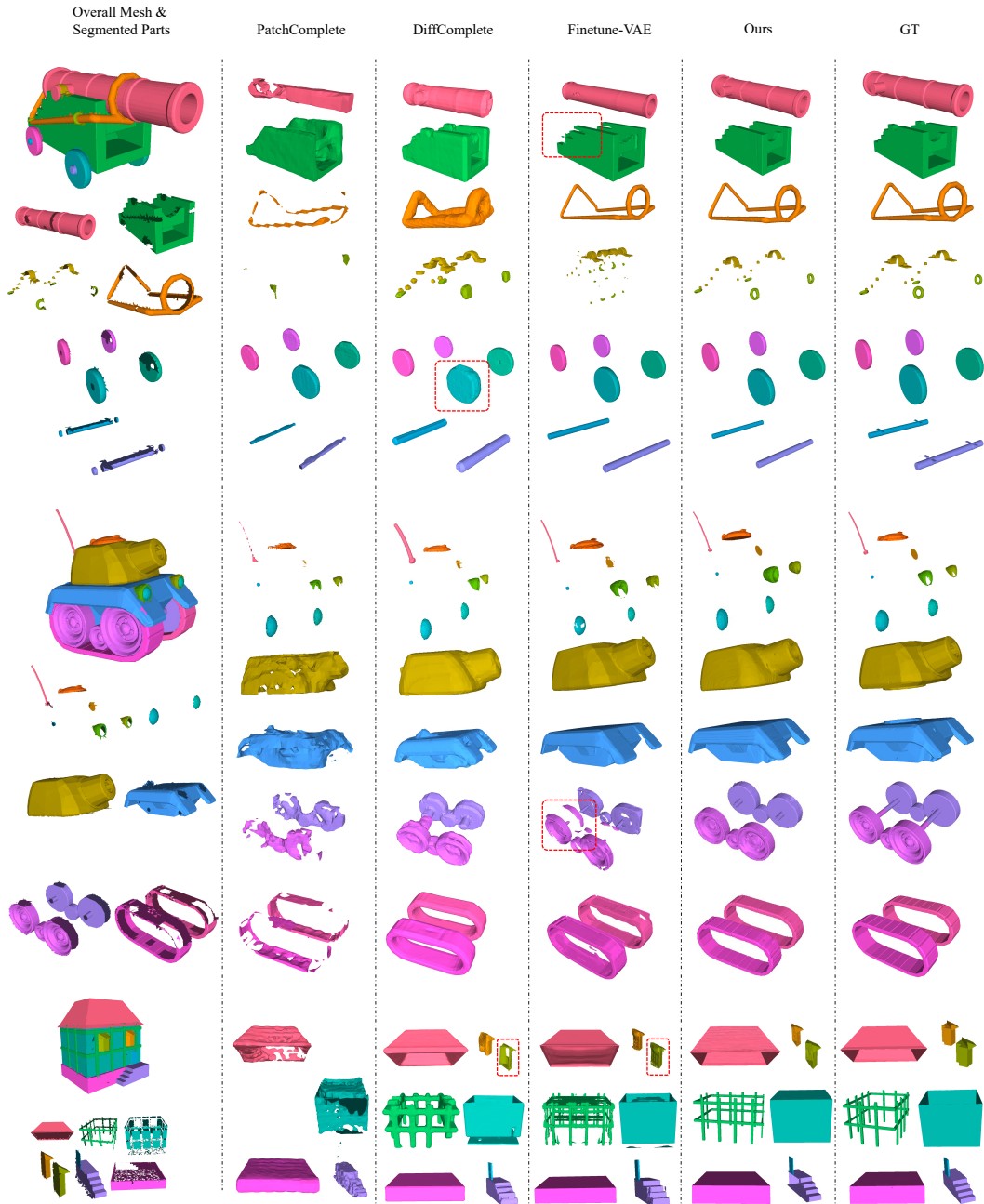

Figure 16: More qualitative results on the PartObjaverse-Tiny dataset.

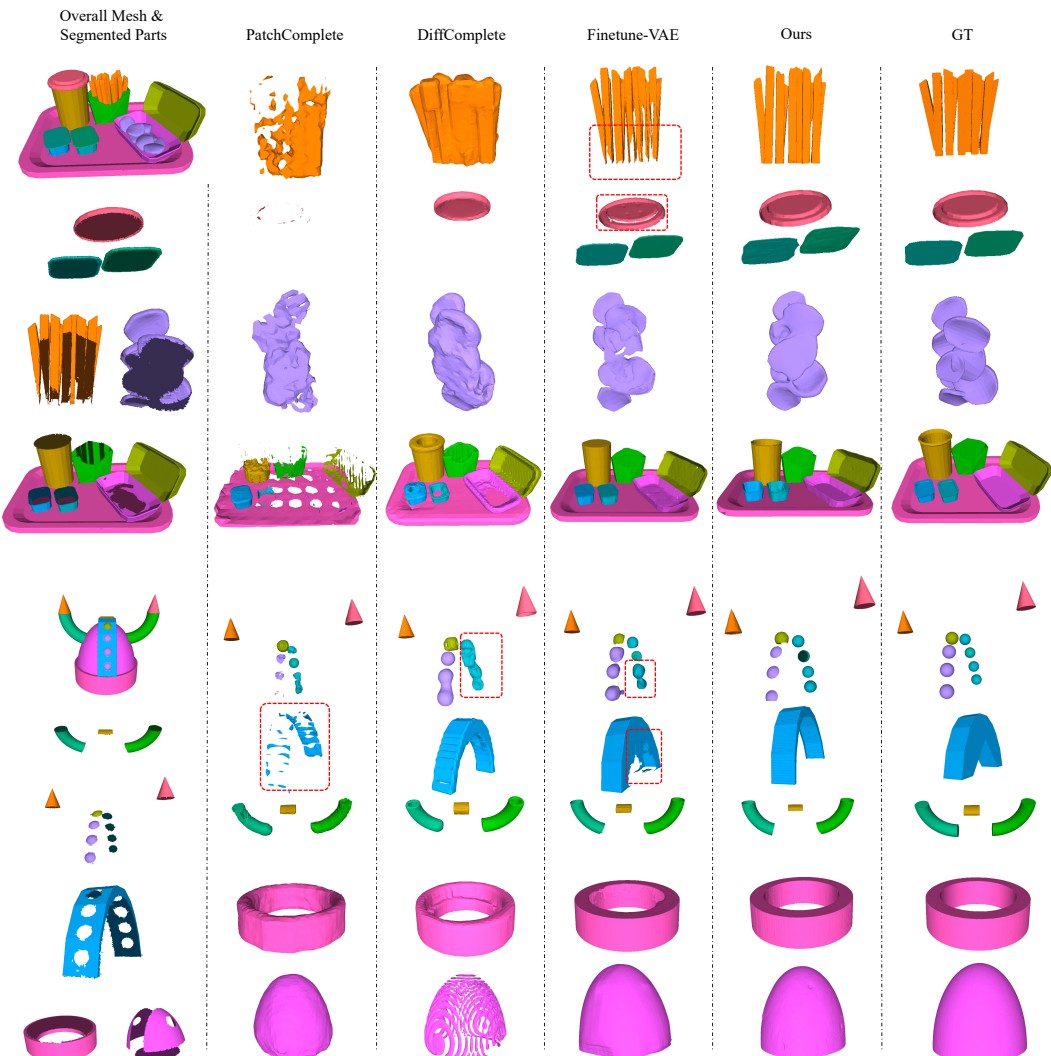

Figure 17: More qualitative results on the PartObjaverse-Tiny dataset.

