# OpenReview forum: "HoloPart: Generative 3D Part Amodal Segmentation"
_ICLR.cc/2026/Conference — ICLR 2026 Poster_

### Official Review · Reviewer_BYbA · 2025-10-30

**Soundness:** 3
**Presentation:** 3
**Contribution:** 3
**Rating:** 6
**Confidence:** 4

**Summary:**

HoloPart introduces a novel diffusion-based model for 3D part shape completion and formally introduces the 3D part amodal segmentation task with two benchmarks (ABO and PartObjaverse-Tiny). It outperforms SOTA shape completion methods on these two benchmarks.

**Strengths:**

1. The 3D part amodal segmentation task and the ability to infer occluded 3D geometry of less complex geometric structures (i.e., Parts) are important for 3D understanding and can be beneficial for broad applications.
2. HoloPart outperforms competitors on the benchmarks on ABO and PartObjaverse-Tiny datasets.

**Weaknesses:**

## Major Weaknesses
1. In the two-stage pipeline, the 3D part segmentation method is crucial for final performance. SAMPart3D is not a very robust method. The segmented part definitions are not in control and often require additional merging for evaluation. Missing parts or incompatible segmented part definitions will significantly impact performance. So I don't think it is a very good initialization for the benchmark evaluation.

## Minor Weaknesses
1. Leveraging 3D generative priors indeed overcomes the limitations of scarce training data. However, it will also introduce their failure modes and may still fail on challenging cases for current 3D shape diffusion models.

**Questions:**

1. Based on weakness 1, given a rendered image of a mesh having 3D part annotations, could we use the 2.5D geometry of this view as the initialization for part completion (only consider visible parts in this view)?  It is a more challenging part-completion scenario. Could you demonstrate HoloPart's performance on this 2.5D part geometry on a small scale (e.g., 20-50 examples)?

2. It would also be valuable to provide additional qualitative and quantitative results of HoloPart on car and airplane categories (68 meshes in total) in the 3DCompat++ [1] dataset to show HoloPart's generalization ability in outdoor objects and with more semantically meaningful part definitions. 3DCompat++ has rendered images, and it would be easy to extract the 2.5D part geometry.

[1] 3DCOMPAT++: A comprehensive dataset for 3D object understanding with fine-grained part annotations

---

> ### Author Response · Authors · 2025-11-21
>
> Thank you very much for taking the time to review and for your support. We try our best to address your questions as follows.
>
> **[W1] Dependence on SAMPart3D & Part Granularity**
>
> - **Precise Completion Evaluation (Tables 1 & 3).** As described in lines 301-305 of the paper, the results in Tables 1 and 3 utilize incomplete masks constructed from Ground Truth (GT). These masks maintain the same granularity as the GT parts. This setup allows us to precisely evaluate the shape completion capability of the various models in isolation, without the interference of inconsistent part definitions or segmentation errors.
>
> - **Full Pipeline Benchmark (Table 4).** To demonstrate performance on the full 3D Part Amodal Segmentation task, we explicitly present results in Table 4 using SAMPart3D integrated with various completion baselines. This establishes a benchmark for the complete pipeline, facilitating fair and direct comparisons for future research on this task.
>
> **[W2] Failure Modes of Generative Priors.**
>
> The reviewer raises a valid point that generative priors can introduce hallucinations or failure modes typical of diffusion models. We would like to highlight two key mitigations in our design:
>
> - **Strong Conditioning.** HoloPart is strictly constrained by the Local Attention module, which anchors the generation to the existing visible surface mesh. This "anchoring" significantly reduces the degree of freedom for hallucination, ensuring the generated part faithfully extends the visible geometry rather than inventing unrelated structures.
>
> - **Contextual Constraints.** The Context-Aware Attention  further constrains the generation space by ensuring the part fits within the global shape context.
>
> **[Q1Q2] 2.5D Geometry Initialization & 3DCoMPaT++ Experiments**
>
> We thank the reviewer for this excellent suggestion. Evaluating on 2.5D geometry from single-view renders is indeed a more challenging and practical scenario. We have added these experiments to the revised paper:
>
> - **Dataset & Setup.** Following your suggestion, we incorporate the 3DCoMPaT++ dataset, specifically selecting the Car and Airplane categories (outdoor objects), along with Bed and Faucet (suitable for 2.5D mask). We project 2D segmentation masks from single-view renderings onto the mesh to obtain 2.5D surface segments (which are highly incomplete and view-dependent) as initialization for HoloPart.
>
> - **Qualitative Results.** As visualized in Figure 6, we illustrate the progression from a single-view input image/mask to the projected 2.5D mask, and finally to the complete 3D part generated by HoloPart. Despite the sparsity of information in this single-view setting, our model successfully reconstructs the complete 3D parts. This validates the robustness and effectiveness of our approach in handling such challenging, limited-view scenarios.
>
> - **Quantitative Results.** We present the comparison with baselines on this 2.5D task in the new Table 2. HoloPart significantly outperforms baselines (PatchComplete, DiffComplete, SDFusion), demonstrating superior generalization to outdoor objects and robust handling of sparse 2.5D input.
>
> | Metric | Category | PatchComplete | DiffComplete | SDFusion | Ours |
> | :--- | :--- | :---: | :---: | :---: | :---: |
> | **Chamfer** $\downarrow$ | car | 0.289 | 0.153 | 0.264 | **0.090** |
> | | airplane | 0.267 | 0.141 | 0.241 | **0.087** |
> | | faucet | 0.258 | 0.125 | 0.232 | **0.076** |
> | | bed | 0.295 | 0.162 | 0.282 | **0.097** |
> | | *mean (instance)* | 0.278 | 0.146 | 0.255 | **0.088** |
> | | *mean (category)* | 0.277 | 0.145 | 0.255 | **0.087** |
> | **IoU** $\uparrow$ | car | 0.247 | 0.382 | 0.323 | **0.545** |
> | | airplane | 0.231 | 0.405 | 0.230 | **0.572** |
> | | faucet | 0.291 | 0.442 | 0.185 | **0.601** |
> | | bed | 0.215 | 0.368 | 0.254 | **0.531** |
> | | *mean (instance)* | 0.245 | 0.401 | 0.246 | **0.558** |
> | | *mean (category)* | 0.246 | 0.399 | 0.248 | **0.562** |
> | **F1-Score** $\uparrow$ | car | 0.314 | 0.485 | 0.406 | **0.635** |
> | | airplane | 0.291 | 0.508 | 0.299 | **0.652** |
> | | faucet | 0.365 | 0.529 | 0.277 | **0.673** |
> | | bed | 0.282 | 0.416 | 0.313 | **0.614** |
> | | *mean (instance)* | 0.312 | 0.485 | 0.321 | **0.641** |
> | | *mean (category)* | 0.313 | 0.484 | 0.323 | **0.644** |
> | **Success** $\uparrow$ | *mean (instance)* | 0.835 | 0.935 | 0.884 | **0.995** |

---

### Official Review · Reviewer_LGMD · 2025-10-31

**Soundness:** 3
**Presentation:** 3
**Contribution:** 3
**Rating:** 6
**Confidence:** 4

**Summary:**

This paper proposes HoloPart, a framework that decomposes a complete 3D object into multiple complete and coherent 3D parts. The pipeline first employs SAMPart3D to obtain semantic mesh segmentation masks, and then feeds each incomplete segmented mesh into the proposed model, which reconstructs the complete 3D part meshes.

The contributions lie primarily in the architectural design of the proposed model and the construction of a new dataset for training and evaluation. The method is compared against state-of-the-art shape completion approaches — PatchComplete, DiffComplete, and SDFusion — and achieves superior performance across Chamfer Distance, IoU, F1-Score, and Success Rate metrics. Ablation studies further validate the effectiveness of the Context-Aware Attention, Local Attention, and the influence of the Guidance Scale.

**Strengths:**

1. The paper tackles a valuable and underexplored problem, focusing on the generation of complete 3D parts rather than full-object completion.
2. The construction of a dedicated dataset for 3D part completion is a useful contribution that can facilitate future research in this direction.

**Weaknesses:**

1. The approach relies heavily on existing 3D part segmentation techniques, which could limit its robustness when segmentation quality is poor.
2. The task formulation assumes the input is a complete object mesh, yet the pipeline includes VAE compression and flow matching in latent space. It is unclear whether the reconstructed meshes remain geometrically consistent with the original object after decoding.

**Questions:**

1. Since the pipeline encodes the 3D mesh via a VAE, performs flow matching in latent space, and then decodes it back, does the reconstructed mesh deviate geometrically from the original input mesh? If so, how do the authors mitigate or correct such deviations?

Things to improve the paper that did not impact the score:
- Table 2: The Success Rate metric is mentioned in the text but not shown in the table — please clarify where it is reported.

---

> ### Author Response · Authors · 2025-11-21
>
> Thank you very much for taking the time to review and for your support. We try our best to address your questions as follows.
>
> **[W1] Robustness to Segmentation Quality**
>
> We appreciate the reviewer raising the concern about reliance on upstream segmentation.
>
> - **Flexible Integration.** Our two-stage framework is designed to be compatible with any surface segmentation method. As shown in Fig. 5, HoloPart works seamlessly with various methods like SAMPart3D, SAMesh, PartField and P3-SAM. We can generate even precise **joint structures**, such as the mortise-and-tenon joints at the robot’s connections shown in the figure.
>
> - **Robustness via Augmentation.** To mitigate errors from poor segmentation (e.g. artifacts), we apply mask augmentation (random noise, erosion, dilation) during training. This forces the model to learn robust shape priors rather than over-relying on perfect input masks. As evidenced in Fig. 15  (noisy mask of glasses) and Fig. 6  (noisy 2.5D masks), HoloPart successfully recovers clean and complete geometries even when input segments are imperfect.
>
> **[W2Q1] Geometric Consistency and VAE Reconstruction**
>
> - **Task Definition.** Our task input is often a "holistic shell" (e.g., scanned or generated meshes) consisting of surface patches, rather than a complete object. The goal is generative completion—inferring missing internal structures and occluded geometries to create solid parts. Therefore, the output is not intended to be a pixel-perfect reconstruction of the input shell, but rather a "completed" version that is geometrically consistent with the visible surface.
>
> - **Preserving Details via Local Attention.** We specifically design the Local Attention mechanism  to encode the geometry of the input segments. This module explicitly attends to the local features of the input surface patch, ensuring that the generated part faithfully aligns with the original visible geometry while "growing" the occluded portions. Our quantitative metrics (Chamfer Distance) in the benchmarks specifically penalize geometric deviations, and our superior performance confirms we maintain high fidelity to the ground truth.
>
> - **Geometry Super-Resolution.** Contrary to losing detail, our method can actually enhance resolution. In standard Whole-Shape VAEs, a fixed number of tokens represents the entire object. In HoloPart, we assign the same number of tokens to represent a single part. This effectively increases the token density per unit area for parts compared to the whole shape. As demonstrated in Fig. 7, this allows for Geometry Super-resolution, where our reconstructed parts exhibit higher geometric fidelity than the original whole-shape VAE reconstruction.
>
> **[Q2] Success Rate metric**
>
> We thank the reviewer for pointing this out. We have updated the Table in the revised manuscript to include the Success Rate metric.

---

### Official Review · Reviewer_WfB3 · 2025-11-01

**Soundness:** 3
**Presentation:** 4
**Contribution:** 3
**Rating:** 6
**Confidence:** 3

**Summary:**

The paper proposes a 2 step procedure to do 3D amodal segementation. Stage 1:  it uses an off-the-shelf approach to segment out the semantically consistent but visible portions of the occluded object.  Stage 2:  uses a local and global context aware diffusion method to  complete the occluded 3d subobject. Additionally they also propose a data pipeline strategy to curate  3D shapes with labeled semantic sub parts and their full (amodal) geometry for training and evaluation

**Strengths:**

1) The central idea of completing the occluded 3d sub part with a local and globally conditioned diffusion model is quite appealing

2) The datapipeline strategy to curate paired data is simple and easy to engineer and would be quite useful for future models.

3) Evaluate their model for various settings and compare to various baselines.

**Weaknesses:**

1) Reliance on segmentation quality of the off-the-shelf model in stage-1. Difficulties would arise due to some type of domain shift, due thin structures or heavy occlusions that prevent the stage 1 model from being able to segment the sub part well.

2) In general two stage training pipelines are a bit clunky as it requires two models to be trained unlike end-to-end trained models.

**Questions:**

I am curious to know about the compute comparison between the proposed model and baselines, in terms of flops/training time/ number of model parameters.

---

> ### Author Response · Authors · 2025-11-21
>
> Thank you very much for taking the time to review and for your support. We try our best to address your questions as follows.
>
> **[W1W2] Reliance on Segmentation Quality & Two-Stage Design**
>
> We thank the reviewer for these insightful comments. We clarify our design choices and robustness strategies as follows:
>
> - **Why Two-Stage?** 3D Part Amodal Segmentation is a challenging task requiring both semantic understanding (segmentation) and geometric inference (completion of occluded/invisible parts). Solving this end-to-end from scratch is extremely difficult due to data scarcity and optimization complexity. Our two-stage design decomposes this into manageable sub-problems. This modularity is a feature: it allows HoloPart to flexibly integrate with any state-of-the-art surface segmentation model. As demonstrated in Fig. 5, our method seamlessly adapts to various upstream models (SAMPart3D, SAMesh, PartField, and P3-SAM), ensuring the framework remains relevant as segmentation models evolve.
> - **Robustness to Segmentation Noise.** To address the concern about reliance on mask quality (e.g., thin structures or artifacts), we implement mask augmentation strategies (random noise, erosion, and dilation) during training. This forces HoloPart to learn robust shape priors rather than over-relying on the input mask. As shown in Fig. 15, our model successfully reconstructs intricate, thin structures like glasses despite imperfect input segments. In Fig. 6, even with noisy 2.5D masks from the 3DCoMPaT++ dataset, HoloPart generates clean and complete part geometries. These results confirm that our generative prior can effectively correct and complete parts even when Stage-1 outputs are imperfect.
>
> **[Q1] Computational Cost Comparison**
>
> We provide the comparison of parameters, training time, and inference cost (TFLOPs each step) below.
>
> | Method  | Resolution | # Params | TFLOPs / Step | Training Time |
> | :---  | :--- | :---: | :---: | :---: |
> | DiffComplete  | Voxel ($32^3$) | 43.1 M | ~0.09 | ~3days  |
> | PatchComplete  | Voxel ($32^3$) | 182.8 M | ~0.36 | ~4days  |
> | SDFusion  | Voxel ($64^3$) | 620.7 M | ~1.3 | ~4days  |
> | **HoloPart (Ours)**  | **Vecset Latent** | **1000 M** | **~4.00** | **~4days** |
>
> Our model employs a Diffusion Transformer (DiT) architecture with 1B parameters to learn a robust 3D generative prior, whereas baselines rely on lightweight voxel-based ($32^3$ or $64^3$) CNNs. While this results in higher computational cost, it is essential for achieving the fine-grained geometry super-resolution (Fig. 7) and zero-shot generalization (Fig. 5) capabilities that low-resolution voxel models cannot match.

---

### Official Review · Reviewer_BTkd · 2025-11-02

**Soundness:** 3
**Presentation:** 3
**Contribution:** 2
**Rating:** 6
**Confidence:** 3

**Summary:**

This paper introduces 3D part amodal segmentation, a new task aimed at decomposing a 3D shape into its complete, semantically meaningful parts, including portions that are occluded. The authors propose HoloPart, a novel diffusion-based generative model that takes incomplete part segments and completes them by leveraging both local part geometry and global shape context.

**Strengths:**

1. The task presented in the paper is well-motivated and practical.
2. The dual local and context-aware attention mechanisms is a good design for balancing fine-grained part detail with overall shape consistency.
3. The presented results look good.

**Weaknesses:**

1. It is unclear how well the model generalizes to novel part compositions not well-represented in the finetuning data, as the completion may be heavily reliant on the part-whole priors learned from the ABO and Objaverse datasets.
2. The method's handling of semantic ambiguity is not discussed. For instance, if a mask incorrectly bridges two distinct semantic parts (like a chair leg and the seat), the model's behavior is unpredictable.
3. The generative completion process can introduce geometric hallucinations that deviate from the original shape's implicit structure. For instance, in Figure 13 (the turkey), the completed parts appear to add new geometry not implied by the original surface, and the final merged object shows significant inter-part overlaps.
4. There is a disconnect between the paper's "amodal segmentation" framing and its core "part completion" contribution. The method is entirely dependent on the quality of the initial segmentation and lacks any mechanism to refine, correct, or handle the noisy and often semantically incorrect outputs of the first stage, thus not fully addressing the end-to-end segmentation problem. Changing the narrative of the paper would make the contribution more aligned.

**Questions:**

Please see the weakness part.

---

> ### Author Response · Authors · 2025-11-21
>
> Thank you very much for taking the time to review and for your support. We try our best to address your questions as follows.
>
> **[W1] Generalization to Novel Part Compositions**
>
> We appreciate the concern regarding generalization. We would like to highlight that HoloPart demonstrates strong zero-shot generalization capabilities beyond the training domain: As shown in Figure 5 (in the revised paper), we evaluate our model on scanned objects from OmniObject3D and generated objects, which possess significantly different distributions from our training data. HoloPart seamlessly integrates with zero-shot segmentation models to produce high-quality parts. In the robot example in Figure 5, HoloPart successfully reconstructs precise mortise-and-tenon joint structures at the connections, demonstrating its ability to infer functional geometry in novel compositions.
>
> **[W2] Handling Semantic Ambiguity**
>
> We address this in the new Figure 10. HoloPart is robust to varying levels of segmentation granularity. If an input mask merges two semantic concepts (e.g., leg + seat), our model treats this as a coarser semantic part and completes the geometry accordingly.
>
> **[W3] Geometric Hallucinations and Inter-part Overlaps**
>
> Inter-part overlaps can exist in the construction of man-made 3D meshes and are present in the ground-truth training data. To prevent unreasonable deviations, our Context-Aware Attention module is explicitly designed to model the relationship between the part and the global shape, keeping the generated geometry consistent with the overall structure.
>
> **[W4] 3D Part Amodal Segmentation**
>
> We thank the reviewer for this thoughtful comment on the paper's framing. We respectfully maintain that "3D Part Amodal Segmentation" is the correct definition for the overarching task we aim to solve, while "Part Completion" is the core technical means to achieve it.
>
> - **Task Significance.** Transforming a "holistic shell" (scanned/generated mesh) into complete, functional parts is a critical yet challenging problem. Solving this end-to-end is currently intractable due to data scarcity and optimization difficulty.
> - **Two-stage Design.** Our two-stage design is a strategic choice for modularity. It allows HoloPart to benefit from the rapid advancements in surface segmentation (e.g., SAMPart3D, SAMesh, PartField, P3-SAM) without retraining.
> - **Robustness Mechanism.** We employ Mask Augmentation (noise, erosion, dilation) during training. This effectively bridges the gap between stage 1 and stage 2. For example, in Figure 15, despite the input mask for the glasses containing significant artifacts and noise, HoloPart successfully recovers clean, distinct geometry.
> - **Comprehensive Evaluation.** To ensure transparency, we evaluate both the second stage in isolation (Tables 1 & 3) and the full Amodal Segmentation pipeline (Table 4), providing a complete picture for future research.

---

### Author Response · Authors · 2025-11-21
**Overall Response**

We sincerely thank the reviewers for their time and constructive feedback. We have revised the paper to address the concerns, highlighting changes in the updated manuscript. Below, we address the common questions raised by the reviewers:

1. **Two-Stage Design.**
3D Part Amodal Segmentation is a challenging task that demands both semantic understanding (segmentation) and geometric inference (completion of invisible parts). This capability is particularly vital for **generated or scanned objects**, which often exist only as **"holistic surface shells"** and are inherently incomplete. Solving this end-to-end is extremely difficult due to data scarcity and optimization complexity. Our two-stage design strategically decomposes this into manageable sub-problems. This modularity is a deliberate design choice, not a limitation: it allows HoloPart to flexibly integrate with any state-of-the-art surface segmentation model. As demonstrated in Fig. 5, our method seamlessly adapts to various upstream models (SAMPart3D, SAMesh, PartField, and P3-SAM), ensuring the framework remains effective as segmentation techniques evolve.

2. **Robustness to Segmentation Noise.**
To address concerns regarding reliance on mask quality (e.g., handling thin structures or artifacts), we incorporate mask augmentation strategies (random noise, erosion, and dilation) during training. This forces HoloPart to learn robust shape priors rather than over-fitting to the input mask. As shown in Fig. 15, our model successfully reconstructs intricate, thin structures (e.g., glasses) despite imperfect input segments. Furthermore, in Fig. 6, HoloPart generates clean and complete part geometries even when given noisy 2.5D masks from the 3DCoMPaT++ dataset, and the quantitative results are shown in Tab. 2. These results confirm that our generative prior effectively acts as a regularizer, correcting and completing parts even when Stage-1 outputs are imperfect.

3. **Generalization Capabilities.**
We provide comprehensive evidence of our model's generalization:

- Performance on 2.5D Completion Tasks: We present a comparison with baselines on the 2.5D completion task in the new Table 2. HoloPart significantly outperforms baselines (PatchComplete, DiffComplete, SDFusion), demonstrating superior generalization to outdoor objects and robust handling of sparse 2.5D inputs.

- Generalization to Novel Part Compositions: We highlight HoloPart's strong zero-shot generalization beyond the training domain. As shown in Figure 5 (revised), we evaluate our model on scanned objects from OmniObject3D and generated objects, which possess significantly different distributions from our training data. HoloPart seamlessly integrates with zero-shot segmentation models to produce high-quality parts. Notably, in the robot example in Figure 5, HoloPart successfully reconstructs precise mortise-and-tenon joint structures at the connections, demonstrating its capacity to infer functional geometry in novel compositions.

---

### Author Response · Authors · 2025-12-02

Dear Area Chair,

We fully understand the significant workload you are facing during this busy decision period. We are truly grateful for the time and dedication you have invested in maintaining the high standards of the review process. We provide this concise comment to assist your evaluation.

**Paper Overview & Consensus.**
The reviewers are unanimous in their positive assessment (All rated 6). There is a strong consensus that the paper tackles a "valuable and underexplored problem" (3D Part Amodal Segmentation) with a "well-motivated" and practical approach. The proposed HoloPart model is praised for its "appealing" core idea and "good design" (balancing local details with global consistency). Furthermore, the introduction of new benchmarks and data pipelines is recognized as a significant contribution that will "facilitate future research" and benefit broad 3D applications.

**Comprehensive Resolution of Concerns.**
We have comprehensively addressed all raised inquiries in our replies:

- **Main Concerns (in "Overall Response"):** We have consolidated our responses to the shared major concerns, including robustness, two-stage design choices, and generalization. This includes detailed clarifications and new experimental results (e.g., 3DCoMPaT++ and zero-shot tests).

- **Specific \& Other Concerns (Individual Replies):** We have addressed all other specific questions raised by each reviewer (such as semantic ambiguity and metric definitions) in their respective individual reply threads.

We believe the revised manuscript and additional experiments firmly support the validity and contribution of HoloPart.

Sincerely,

The Authors

---

### Meta-Review · Area_Chair_xAfA · 2026-01-06

**Summary:**

The authors propose a novel task called 3D part amodal segmentation. Inspired by 2D amodal segmentation for images, 3D amodal segmentation refers to decomposing a given 3D shape into parts while completing unseen regions. For this task, the authors combine an existing 3D segmentation method with a 3D shape completion method based on a diffusion model. Their framework, trained on ABO and PartObjaverseTiny, achieves the best performance compared to other shape completion methods and shows good generalizability to unseen shapes.

**Reviewer Concerns:**

Reviewers raised concerns about:
        1.        generalizability to unseen shapes
        2.        semantic ambiguity
        3.        hallucination in the shape completion results
        4.        error accumulation in the two-stage framework (errors in the initial segmentation can lead to poor completion results)
        5.        preserving the given shape during completion

For (1), the authors provide some qualitative results on unseen 3D shapes. Concern (4) seems inherent to a two-stage pipeline, and (2), (3), and (5) are also common issues in segmentation and completion methods in general.

One issue that needs to addressed in the final version is the framing of the paper: while the authors present the work as proposing a new task, the method is essentially a combination of 3D segmentation and 3D shape completion. Moreover, the main technical novelty appears to lie only in the 3D shape completion component. The authors also mainly compare their framework against other 3D shape completion methods. Given this, it is important to discuss the advantages and differences of this work against standard shape completion benchmarks.

**Reviewer Scores:**

All reviewers gave a score of 6. The AC and discussed the the SAC. The SAC decides to accept the paper due to its technical contributions. It is recommended that the authors address the issue related to how to position this paper.

---

### Decision · Program_Chairs · 2026-01-26

Accept (Poster)